# Role of Branched-Chain Amino Acid Metabolism in Type 2 Diabetes, Obesity, Cardiovascular Disease and Non-Alcoholic Fatty Liver Disease

**DOI:** 10.3390/ijms23084325

**Published:** 2022-04-13

**Authors:** Paola Cuomo, Rosanna Capparelli, Antonio Iannelli, Domenico Iannelli

**Affiliations:** 1Department of Agricultural Sciences, University of Naples “Federico II”, Via Università, 100-Portici, 80055 Naples, Italy; paola.cuomo@unina.it; 2Department of Digestive Surgery, Université Côte d’ Azur, F-06108 Nice, France; iannelli.a@chu-nice.fr; 3Centre Hospitalier Universitaire de Nice-Digestive Surgery and Liver Transplantation Unit, Archet 2 Hospital, 151 Route de Saint Antoine de Ginestiere, F-062024 Nice, France; 4Inserm, U1065, Team 8 “Hepatic Complications of Obesity and Alcohol”, F-062024 Nice, France

**Keywords:** branched-chain amino acids, bariatric surgery, adiponectin

## Abstract

Branched-chain amino acids (BCAAs) include leucine, isoleucine, and valine. Mammalians cannot synthesize these amino acids de novo and must acquire them through their diet. High levels of BCAAs are associated with insulin resistance; type 2 diabetes; obesity; and non-metabolic diseases, including several forms of cancer. BCAAs—in particular leucine—activate the rapamycin complex1 mTORC1, which regulates cell growth and metabolism, glucose metabolism and several more essential physiological processes. Diets rich in BCAAs are associated with metabolic diseases (listed above), while diets low in BCAAs are generally reported to promote metabolic health. As for the dysregulation of the metabolism caused by high levels of BCAAs, recent studies propose that the accumulation of acyl-carnitine and diacyl-CoA in muscles alters lipid metabolism. However, this suggestion is not broadly accepted. On clinical grounds, pre- and post-operative metabolic profiles of candidate patients for bariatric surgery are being used to select the optimal procedure for each individual patient.

## 1. Introduction

Branched-chain amino acids (BCAAs) include leucine, isoleucine and valine. Mammalians cannot synthesize BCAAs de novo and must therefore acquire them through their diet. However, there is evidence that bacteria present in the human microbiota can synthesize BCAAs [1]. This class of amino acids acts as biomarkers of several diseases, such as type 2 diabetes (T2D), obesity (OB) cardiovascular diseases (CVD) and several forms of cancer, while their metabolites can regulate gene expression and the epigenome [2]. BCAAs are also important as components of the diet. Leucine is the prevalent component of protein; isoleucine and valine provide carbon for the synthesis of glucose; and the amino acids derived from the BCAA catabolism feed the tricarboxylic acid cycle and provide energy to cells. Intracellular amino acids—in particular leucine—activate the rapamycin complex1 (mTORC1), which regulates cell growth and metabolism; integrates signals from nutrients and energy sources; promotes cell growth when resources are sufficient; and promotes catabolism when the body is starved [3]. Ordinarily, amino acid degradation takes place in the liver, but the branched-chain aminotransferase (BCAT) (the enzyme catalyzing transamination of BCAAs) is not expressed in the liver. Consequently, BCAAs enter directly from the gut into circulation. This in part explains why the blood level of BCAAs can rapidly change, depending on the diet and the rate of protein synthesis or degradation. The level of BCAAs influences many physiological processes, such as mitochondrial biogenesis, energy metabolism, inflammation and glycolysis [4]. This review focuses mainly on the role of BCAAs in T2D and obesity.

## 2. Properties of BCAAs

Obesity is a worldwide health problem. In the United States, two-thirds of the population are overweight or obese [5]. Numerous studies have concluded that ‘‘a calorie is not just a calorie’’ [6]; put simply, all calories have the same amount of energy, but, in our body, different foods go through different biochemical pathways. Often, but not always, a diet rich in BCAAs is reported to be associated with high blood levels of BCAAs, in IR, obesity, T2D [7] and CVDs [8]. Conversely, a diet low in BCAAs is often reported to promote metabolic health and longevity in mammalian species, including mice and humans [9]. Further studies in rats have shown that a diet low in BCAAs is associated with insulin sensitivity and reduced fat accumulation [10].

The majority of the studies carried out to date have investigated the role of BCAAs grouped together. However, more recent studies have demonstrated that each BCAA influences metabolism differently. The three BCAAs activate mTORC1, a key regulator of metabolism, differently [11]; modify different proteins post translationally [12]; and produce different intermediates and final catabolic products [13]. In line with the above studies, high blood levels of leucine have been found to be associated with decreased all-cause mortality [14]. In obese mice, leucine and isoleucine both correct IR and favor browning of white adipose tissues (WAT) [15]. In patients with obesity, dietary supplementation with leucine favors weight loss, reduces WAT inflammation and increases mitochondrial function [16]. At present, leucine supplementation is receiving much interest for the treatment of T2D and obesity. A large study in this field [17] concluded that although central leucine injection decreases food intake, this effect is not well reproduced. Methods that can better dissect the multiple factors (diet, gut microbiome and physical activity) influencing BCAA metabolism are being developed [18]. Thus, it is reasonable that leucine will soon find its therapeutic application.

Patients with T2D display high levels of BCAAs in the skeletal muscle and activation of mTORC1/S6K1 [19,20]. Under this condition, the patient displays impaired insulin signaling by attenuating *P13K/Akt*, leading to IR [20]. This circumstance can be explained assuming that starved myotubes—in the presence of leucine—inhibit glucose uptake and consequently insulin signaling by increasing *S6K1* phosphorylation. The same occurs with *KIC*, a leucine metabolite, confirming that the inhibition of insulin activity is associated with an increase in *S6K1* phosphorylation. The effect of this metabolite (*KIC*) is reduced in BCAT2 depleted cells, suggesting that the metabolite is again converted into leucine to continue inhibiting glucose from entering into cells. In line with these results, skeletal muscles deprived of the enzyme *BCKDH* display reduced glucose uptake, without changing the phosphorylation state of *S6K1*, suggesting the presence of an alternative mechanism of insulin activity inhibition [21]. This conclusion is supported by the evidence that increased levels of the metabolite 3-hydroxiisobutirate (a valine metabolite) contribute to insulin resistance by favoring the accumulation of fatty acids in skeletal muscles. In line with this finding, BCAA restriction in Zucker rats improves insulin sensitivity in skeletal muscle and, at the same time, favors fatty acid oxidation [13]. However, although high levels of BCAAs lead to IR, athletes who consume diets rich in BCAAs do not display IR [22]. This apparent contradiction is explained if we recall that athletes consume the high level of BCAAs present in their diet and consequently do not accumulate BCAAs.

## 3. BCAAs and Metabolic Diseases

Chronic high blood levels of BCAAs are associated with obesity, T2D [7] and CVD [8]. In individuals with obesity, high levels of BCAAs, tyrosine and phenylalanine, and low levels of glycine, are associated with IR [23]. The metabolic analysis of blood samples from obese and insulin-resistant versus lean and insulin-sensitive subjects confirmed that high levels of BCAAs are strongly associated with IR [18].

It is known that fatty acids and their metabolites cause IR and T2D [24]. However, studies in rodents have shown that adding BCAAs to a standard chow (SC) diet does not influence the level of insulin. However, adding BCAAs to a high-fat diet (HFD) induces obesity and IR [7]. These data indicate that IR induction requires the presence of both BCAAs and lipids. Recent studies provide experimental evidence of a close interaction between BCAAs, diet and lipid metabolism. A study conducted on an animal model of obesity reports that the accumulation of acyl-carnitine and acyl-CoA in muscles is attributable to fatty acids in complete oxidation and mitochondrial overload [25]. In this context, the excess of BCAAs leads to increased mitochondrial activity, which alters lipid metabolism. In line with these data, it has been observed that a diet low in BCAAs corrects lipid metabolism [26]. This process involves the activation of the general control non-derepressable 2 (*GCN2*) gene, which inhibits the expression of lipogenic genes and protein synthesis; re-establishes the physiologic level of glycine in the muscle; and favors the synthesis of acyl-glycine, which—discharged through urine—reduces the acyl-CoA levels (Figure 1).

Accumulation of incompletely esterified lipids (acyl-carnitine and diacyl-glycerol) in the skeletal muscle of insulin-resistant rodent models is consistent with the above results [10,13]. Based on these findings, it has been proposed that IR can be attributed to an excess of BCAA catabolites and the accumulation of partially esterified lipids in the muscle, which leads to lipotoxicity, glucose intolerance, IR and, finally, T2D [7].

Metabolomics has also demonstrated that high blood levels of BCAAs predict the development of T2D more than 10 years before it develops. When the first blood samples were collected, none of the participants in the study showed signs of T2D or IR [27]. An independent study confirmed these results [28]. Bariatric surgery induces a rapid decline in BCAAs and aromatic amino acids (AAAs) [29]. Instead, high maternal levels of BCAAs, AAAs and their metabolites during pregnancy affect fetal adiposity [30].

A Mendelian randomization study concluded that BCAAs cause IR [31]; another study reached a contradictory conclusion, namely, that IR causes high levels of BCAAs [32]. It has also been proposed that hypothalamic IR impairs plasma BCAA metabolism in T2D and Obesity; that is, plasma BCAAs are markers of hypothalamic IR, rather than direct cause of peripheral IR [33]. Two large human studies claimed that high dietary BCAAs are associated with T2D [34,35]; conversely, a third study concluded that high levels of dietary BCAAs reduce the risk of T2D [36]. Several studies on the role of BCAAs carried out in rodents also led to discordant results [23], which have been imputed to the use of all three BCAAs in some studies and leucine alone in others [23].

The prevalence of non-alcoholic fatty liver disease (NAFLD) is increasing worldwide as consequence of the obesity epidemic. The most common phenotype of NAFLD—liver steatosis—exhibits hepatic accumulation of triglycerides. Acetic acid is a short-chain fatty acid (SCFA) produced from cellulose by several bacterial species present in the gut microbiota. At present, little is known on the effect of BCAAs on NAFLD. The objective of the study that we are referring to was to ascertain whether a high-fat diet supplemented with BCAAs in rats would influence acetic acid secretion. The authors found that implementing a high-fat diet with BCAAs in rats, significantly promoted the growth of *Ruminococcus (R.) flavefaciens* and portal acetic acid production while reducing the hepatic fat accumulation in the rats and the expression level of the lipogenesis genes *FAB* and *ACC* (Figure 2).

The present case and the one described earlier help resolve whether BCAAs are friends or foes; the answer is both, depending on the context. As mentioned previously, BCAAs are involved in many vital physiological processes [37]. This circumstance, at least in part, explains why their roles are sometimes contradictory and still not completely understood. Recent studies propose that the opposing activities of BCAAs depend on several factors, which future studies should consider: the state of endogenous metabolism, in addition to exogenous metabolism (diet) [38], and the human gut, which influences the host serum metabolome [39,40,41,42].

In conclusion, many independent studies have convincingly demonstrated that high levels of circulating BCAAs are strongly associated with IR, T2D, obesity, CVD and NAFLD (Figure 2 and Figure 3) [43,44,45]. However, the presence of BCAAs in many metabolic pathways and their essential role in several physiological processes—protein synthesis, modulation of glucose metabolism, regulation of appetite and adiposity [19,20]—make it very challenging to establish whether BCAAs are the cause or biomarkers of IR and IR-associated disease [20].

## 4. Altered Expression of BCAA Metabolism Genes

Patients with T2D exhibit higher plasma levels of BCAAs than those of controls. This condition is attributed to insulin resistance [21]. A recent study found that the BCAA metabolite 3-hydroximesobutyrate secreted from muscles causes insulin resistance [13], another sign of the complexity of insulin resistance’s origin. Patients with early-onset T2D (displaying the disease before 30 years of age) and patients with classical-onset T2D (displaying the disease after 50 years of age) both exhibit reduced expression levels of genes regulating BCAAs. In the study we are describing, gene expression was detected by microarray, in muscles from patients with T2D and controls [13]. In patients with classical-onset T2D, there were 790 dysregulated genes: 366 upregulated and 424 downregulated. In patients with early-onset T2D, there were 449 dysregulated genes: 268 upregulated and 181 downregulated. The higher number of dysregulated genes detected in patients with classical-onset T2D suggests that gene dysregulation might be influenced by the disease time-span. Patients with classical- and early-onset T2D were both enriched in BCAA metabolic pathways. To study the origin (genetic or epigenetic) of altered gene expression in more depth, the same patients were analyzed for the methylation status of genes involved in BCAA synthesis. DNA methylation suppresses gene expression by controlling the access of transcription factors to the chromatin. Patients with early-onset T2D displayed reduced expression of *BCAT2* in skeletal muscle. The patients with classical T2D displayed reduced expression of *BCAT2* and *BCKDHB*. In conclusion, the above data indicate that the genes regulating the expression of BCAAs in skeletal muscles are expressed differently, both in patients with early- and classical-onset T2D.

## 5. Factors Interfering with BCAA Functions

### 5.1. The Microbiota Can Alter the BCAA Level of the Host

Separate groups of germ-free mice were transplanted with fecal microbiota from four female twin pairs discordant for obesity (enrolled in the Missouri adolescent female twin study) [46]. Cohousing in the same cage mice transplanted with the microbiota from the obese twin (Ob) with mice transplanted with the microbiota from the lean co-twin (Ln) precluded the development of obesity in mice transplanted with the microbiota from the Ob twin. The same mice displayed (1) increased abundance of Bacteroides, known to prevail in the microbiota of lean mice [1]; (2) increased expression of the genes involved in the degradation of BCAAs; and (3) low levels of circulating BCAAs and aromatic amino acids, as observed in Ln–Ln control mice [47].

There is solid evidence that bariatric surgery is the most effective method for treating T2D [48,49]. Bariatric surgery known as side-to-side jejunoileal bypass plus proximal loop ligation (SSJBL) has been used with success to treat obesity and T2D [50]. It is also known that the change in gut microbiota that follows bariatric surgery contributes to weight loss and glucose homeostasis.

GK rats (used to study non-overweight T2D cases) were randomly distributed into two groups: one group of rats underwent SSJIBL and the other served as the control group. Six weeks after surgery, GK rats, which had undergone SSJIBL, displayed significantly decreased levels of the following variables: body weight, fasting blood glucose and serum lipid levels. Furthermore, the operated rats displayed a significantly increased abundance of *Escherichia coli* and *Ruminococcus gnavus* and a decreased presence of *Prevotella copri* in their gut microbiota. The same group of rats also exhibited a significantly improved β-cell function, glucose tolerance and insulin resistance. Finally, the levels of serum glycine, histidine and glutamine were increased, while that of BCAAs was reduced [39]. The authors—based on the coincident reduced level of BCAAs and of the abundance of *Prevotella copri*, which induces the expression of genes involved in the degradation of BCAAs [40]—convincingly propose that the decrease in *Prevotella copri* abundance improves IR by reducing the level of serum BCAAs.

The two studies described above show that microbiota can alter the circulating BCAA level of the host. However, both studies leave open the question of whether the synthesis of BCAAs is directly attributable to one bacterial species present in the microbiota of the obese donor or indirectly attributable, through one of their metabolites, by inducing the expression of the host genes involved in the synthesis of BCAAs [40,51].

### 5.2. BCAA Levels Are under the Influence of Adiponectin

The reduced catabolic activity of BCAAs caused by the branched-chain α-keto acid dehydrogenase (BCKD) often accounts for the high levels of plasma BCAAs in patients with T2D or other metabolic diseases [52]. Several hormones are involved in the downregulation of BCAA catabolism [26]. The mitochondrial phosphatase 2C (PP2Cm)—a BCKD phosphatase that upregulates BCKD—is markedly downregulated in ob/ob (leptin-deficient) mice with T2D. In adiponectin knockout (APN^−/−^ mice fed with a high-fat diet (HI) (HD-fed APN^−/−^), *PP2Cm* expression was downregulated, while BCKD kinase (BDK)—which inhibits BCKD activity—was markedly upregulated. At the same time, circulating BCAAs and BCKAs were markedly upregulated. The same mice—treated with APN—returned to the original levels of PP2Cm, BDK, BCKD, BCAAs and BCKAs. Furthermore, the increased activity of BCKDs induced by APN treatment was partially inhibited in APN^−/−^ mice. These results show that APN regulates *PP2Cm* and the blood level of BCAAs; on clinical grounds, they suggest that APN may pharmacologically improve BCAA catabolism in patients with T2D [52].

Furthermore, APN treatment can in part reduce the high levels of BCAAs detected in HD-fed APN^−/−^ mice [53]. This result suggests that molecules other than APN present in HD- APN^−/−^ mice can replace APN under normal physiological conditions, while APN becomes essential under stress conditions (HD or T2D) [53]. In other words, APN improves BCAA catabolism in patients with T2D, inducing both BDK and PP2cm in the opposite direction (Figure 4).

### 5.3. Glucose Reduces BCAA Catabolism in Patients with T2D

T2D leads to chronic hyperglycemia, insulin resistance and high levels of BCAAs. At present, it is not clear whether BCAAs are biomarkers of T2D or contribute in part to the disease, or whether the altered metabolism associated with T2D influences the metabolism of BCAAs. The majority of the studies aimed at solving this issue were conducted in people who fasted for 12 h. However, this state induces protein degradation in skeletal muscle to sustain gluconeogenesis (the main source of glucose during a fasting period), a condition that could potentially hide the altered metabolism of BCAAs. A recent study adopted a new approach to clarify this point [54]. The participants in the study—32 men with normal glucose tolerance and 29 with T2D—were asked to fast for 12 h. Blood samples and biopsies from the vastus lateralis muscle were taken from all the participants before and after the oral glucose tolerant test (one dose of 73 gr of glucose per participant). The patients with T2D displayed levels of BCAAs that were 10% (in the plasma) and 13% (in the skeletal muscle) higher than those of the glucose-tolerant participants. Furthermore, the levels of branched-chain keto acids were reduced by 37–56% in the glucose-tolerant participants while remaining unchanged in patients with T2D. Finally, the peroxisome proliferator-activated receptor gamma coactivator 1 alpha (*PGC-1α*) gene—which regulates the expression of the BCAA catabolism genes—was downregulated in patients with T2D.

Altogether, the study suggests that (1) high levels of BCAAs and altered transcription of the BCAA genes are induced by the metabolic disorder associated with T2D; (2) the oral glucose tolerant test may be used to detect altered metabolism of BCAAs; (3) targeting the *PGC-1α* gene may improve the deranged metabolism of BCAAs.

## 6. BCAAs and Bariatric Surgery

Many studies show that the levels of most amino acids (including BCAAs) decrease following bariatric surgery, while a few (glycine and serine) increase independently of the surgical procedure used. At present, bariatric surgery is the most effective means to achieve sustained weight loss and remission or improvement of obesity-related co-morbidities, including T2D, blood hypertension, NAFLD and CVD. Although bariatric surgery is effective in inducing body weight loss and T2D remission [55], 10–20% of patients undergoing bariatric surgery regain weight at various points after surgery. Most of the time, weight regain is associated with the recurrence of previously remitted co-morbidities. At present, pre- and post-operative metabolic profiles of candidate patients for bariatric surgery are used to predict weight loss and metabolic disorders, including T2D remission after surgery. Analogously, the metabolic profiles of patients who have undergone different bariatric procedures might be used to construct a metabolic platform helping the surgeon to decide which is the best bariatric procedure for each individual patient. As a model, we suggest a metabolic platform identifying metabolic predictors of all-cause mortality [14]. Nowadays the most used bariatric procedures worldwide are the sleeve gastrectomy (SG) and the Roux-en-Y gastric bypass. The former consists of a vertical gastrectomy over an intraluminal orogastric boogie that transforms the stomach into a long and narrow tube. The latter includes a small proximal gastric pouch of around 20–30 mL and a long Roux-en-Y loop that allow the food to bypass most of the stomach, the duodenum and a variable part of the proximal bowel. Other procedures, such as biliopancreatic diversion, include a much longer intestinal bypass.

The major differences between Roux-en-Y gastric bypass (RYGB) and sleeve gastrectomy (SG) concern amino acids and gut-microbiota metabolites [56]. Phospholipids are essential components of the mammalian cell membrane, with sphingomyelin (SM) being the most abundant phospholipid. SG induces an increase in sphingolipids and no change in bile acids, while biliopancreatic diversion (BPD) induces a decrease in sphingolipids and no change in biliary acid (Bas) levels [57]. Decreased levels of BCAAs are associated with weight loss after SG and T2D remission after SG or RYGB [29]. Bariatric procedures also influence the metabolites secreted from the microbiota. The level of p-cresol—a metabolite of phenylalanine and tyrosine—increases after SG, while trimethylamine-N-oxide (TMAO), secreted from the gut microbiota, increases after RYGB but not after SG [58]. This result may be ascribed to the intestinal bypass, which creates conditions favorable for bacteria (*E. coli* and *Pseudomonas*) secreting TMAO [59]. Altogether, these studies suggest that pre-operative metabolic profiles might soon be used as prognostic biomarkers for weight loss and T2D remission. Pre-operative high levels of long-chain fatty acids have been reported to be associated with post-operative T2D remission.

## 7. Conclusions

Calorie-dense Western diets are one of the major factors contributing to the worldwide rising numbers of patients with T2D and obesity [60]. Western diets have been reported to consist of >20% BCAAs [61]. High levels of BCAAs are associated with many metabolic diseases, including IR, T2D and obesity. Indeed, the association between BCAAs and IR is more robust than that between IR and lipids. In addition, high levels of BCAAs predict well in advance the development of T2D. However, whether the high levels of BCAAs are cause or biomarkers of the above diseases is not known. This limitation—at least in part—is attributable to the presence of BCAAs in multiple pathways [62] and to their essential role in many physiological processes, such as the regulation of glucose metabolism, and insulin secretion balancing, independently from mTORC1 [33]. Recent studies have shown that pre- and post-operative metabolic profiles of candidate patients for bariatric surgery may predict which type of surgery may give better results (T2D remission and lasting weight loss) [63]. A metabolite platform that helps to identify the lowest risk of all-cause mortality of each individual already exists and has been validated [14]. An analogous platform could push further the above target and identify the types of bariatric surgery associated with the lowest risk of all-cause mortality of each patient: one small step towards precision medicine. Finally, altered metabolism of BCAAs—along with the analysis of the relevant genes involved—helped to identify, among patients with breast cancer, those who could better respond to a specific therapy [64]. The approach—extended to metabolic diseases—might yield equally interesting results. Finally, BCAAs have been studied in several muscle wasting diseases for nearly 50 years, but there is no consensus yet as to their therapeutic utility. This condition and the case of athletes (mentioned above) demonstrate how BCAA metabolism is largely influenced by the context. For several diseases (T2D, obesity and CVD), age is the major risk factor. Understanding how BCAA catabolism changes with aging may help us to better understand and prevent these diseases.

## Figures and Tables

**Figure 1 ijms-23-04325-f001:**
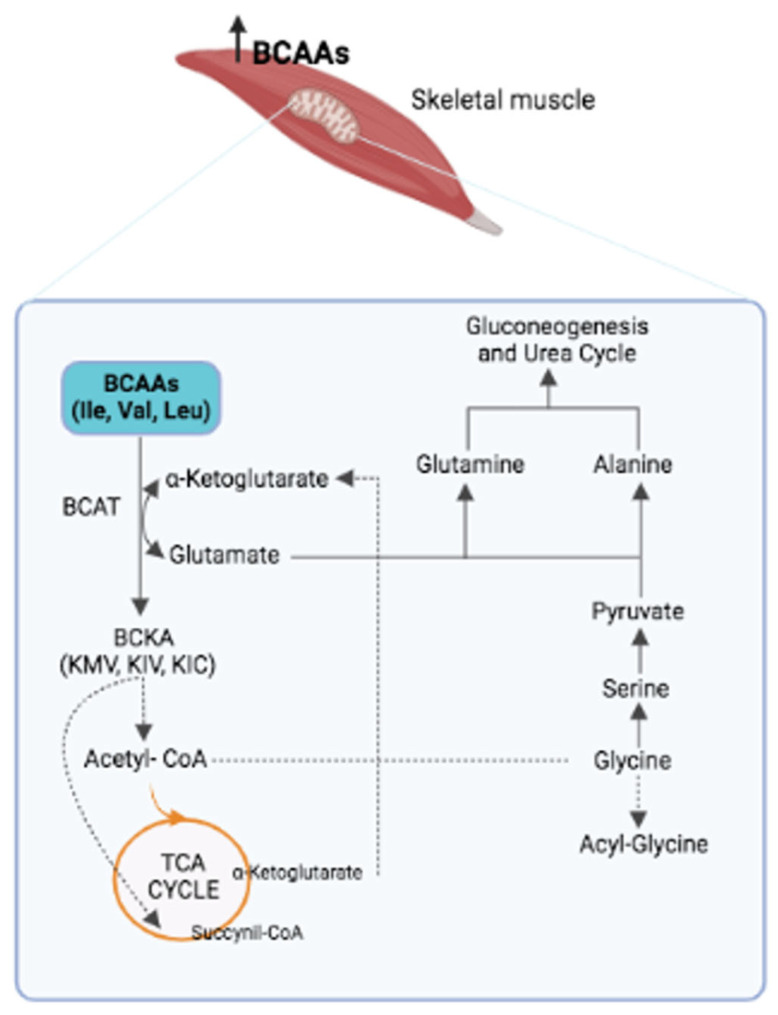
The first step in the catabolism of BCAAs is their transamination by the BCAT enzyme, which leads to three branched-chain keto acids (BCKAs): α-ketoisocaproic acid (KIC), 2-oxoisovalerate (KIV) and α-keto-β-methyl-n-valeric acid (KMV). At the same time, the amino group of each BCAA is transferred to the α-ketoglutarate to produce glutamate as nitrogen source, which, in turn, is converted into glutamine (by the glutamine synthase) and alanine (by the alanine transaminase), which enter the urea cycle and gluconeogenic pathways. This metabolic process leads to a reduced presence of glycine, which is transformed into serine, pyruvate and, finally, into glutamine and alanine. This process explains why high levels of BCAAs in IR are associated with low levels of glycine.

**Figure 2 ijms-23-04325-f002:**
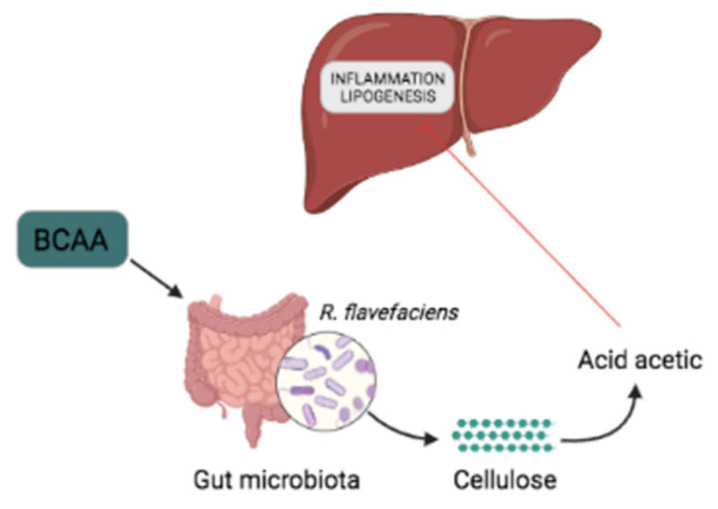
BCAAs alleviate NAFLD. BCAA supplementation favors the abundance of *R. flavefaciens.* and acetic acid synthesis, which reduces liver inflammation and the expression of lipogenesis genes (*FAB* and *ACC*).

**Figure 3 ijms-23-04325-f003:**
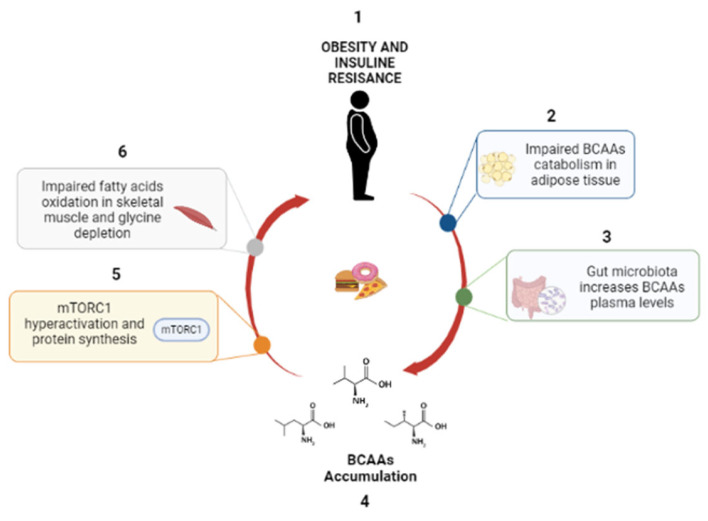
Role of BCAAs in OB. (1) OB and IR alter the catabolism of BCAAs; (2) gut microbiota implement BCAA levels, (3) causing BCAA accumulation (4), mTORC1 activation (5) and inhibition of lipolysis (6).

**Figure 4 ijms-23-04325-f004:**
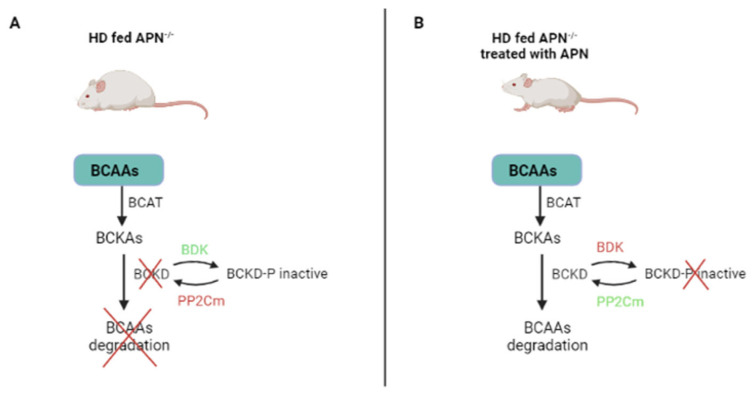
APN regulates BCAA catabolism. (**A**) In the absence of APN, BDK is upregulated and inhibits BCAA catabolism. (**B**) In the presence of APN, BDK is downregulated and BCAAs are degraded. HD: high-fat diet; APN: adiponectin; APN^−/−^ mice: adiponectin knockout mice; BCAAs: branched-chain amino acids; BCKAs: branched-chain keto acids; BCKD: branched-chain keto acid dehydrogenase (induces BCAA catabolism); *PP2Cm*: mitochondrial phosphatase 2C (activates BCKD and induces BCAA catabolism); BDK: branched-chain keto acid dehydrogenase kinase (inactivates BCKD and inhibits BCAA catabolism).

## Data Availability

Not applicable.

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
