# Peer review of "Role of Branched-Chain Amino Acid Metabolism in Type 2 Diabetes, Obesity, Cardiovascular Disease and Non-Alcoholic Fatty Liver Disease"

_ijms, 2022, doi:10.3390/ijms23084325_

Round 1
Reviewer 1 Report
The manuscript by Cuomo et al is of interest and overall it is well written. The references used as the source of information presented are adequate and the information is well organised.
However this reviewer has a major concern: the authors must consider all the data from literature and give a critical appraisal in this very important issue. Which are the most important research and why? are all studies of high validity? enough evidence or unpowered studies and just opinions? what is the authors criticism?
Author Response
The manuscript by Cuomo et al is of interest and overall it is well written. The references used as the source of information presented are adequate and the information is well organised.
However this reviewer has a major concern: the authors must consider all the data from literature and give a critical appraisal in this very important issue. Which are the most important research and why? are all studies of high validity? enough evidence or unpowered studies and just opinions? what is the authors criticism?
Fully aware of the complexity of the issue (BCAAs) and of the involved diseases (all polygenic), the authors decided to screen the literature first individually and then collectively. The different competences of this collective brain helped to better appreciate the literature (please see below) and in some instances to dissect genetics from the environment (gut microbiome). Our approach looks to us very close to the one you would wish. However, the authors acknowledge that they did not clearly explain that the incongruent results about BCAAs reflect their complexity. We have repaired this mistake; please see page 2, lines 66-73.
The 67 articles screened include one article from each of following journals: JAMA, IAMA Pediatr, NEJM, and Metabolism; two from: Science, Nat Med, Diabetologia, Am J Phys Cell Physiol; 10 from Cell Metab or Mol Metab.
Reviewer 2 Report
This review by Paola Cuomo et al had major problems as follow
- The point of view in the manuscript was biased.
The authors declared that “Diets rich in BCAAs are associated with metabolic diseases, while diets low in BCAAs promote metabolic health.
However, a large number of studies showed the beneficial effects of BCAAs on obesity and related metabolic diseases.
For examples
“Leucine and isoleucine have similar effects on reducing lipid accumulation, improving insulin sensitivity and increasing the browning of WAT in high-fat diet-induced obese mice”, Ma et al., Food & Function, 2020
“Leucine in Obesity: Therapeutic Prospects”, Yao et al., Trends in Pharmacological Sciences, 2016
“Leucine as a pharmaconutrient in health and disease”, Van Loon et al., Current opinion in clinical nutrition and metabolic care, 2012
“Reviewing the Effects of l-Leucine Supplementation in the Regulation of Food Intake, Energy Balance, and Glucose Homeostasis”, Pedroso et al., Nutrients, 2015
- The manuscript was not well-structured.
Part 5, “How high levels of BCAA influence the metabolic homeostasis of the body” should be involved in Part 2 “Adverse effects of isoleucine and valine”
NAFLD is also a metabolic disease, Part 9 should be involved in Part 3
Part 4 “Altered expression of BCAA metabolism genes”, it is improper that these contents presented as an independent part.
Part 7, 8 should be Part 6.2 and 6.3.
- The title should be revised. CVD and NAFLD were mentioned in the manuscript.
Author Response
This review by Paola Cuomo et al had major problems as follow
- The point of view in the manuscript was biased.
The authors declared that “Diets rich in BCAAs are associated with metabolic diseases, while diets low in BCAAs promote metabolic health.
However, a large number of studies showed the beneficial effects of BCAAs on obesity and related metabolic diseases.
For examples
“Leucine and isoleucine have similar effects on reducing lipid accumulation, improving insulin sensitivity and increasing the browning of WAT in high-fat diet-induced obese mice”, Ma et al., Food & Function, 2020
“Leucine in Obesity: Therapeutic Prospects”, Yao et al., Trends in Pharmacological Sciences, 2016
“Leucine as a pharmaconutrient in health and disease”, Van Loon et al., Current opinion in clinical nutrition and metabolic care, 2012
“Reviewing the Effects of l-Leucine Supplementation in the Regulation of Food Intake, Energy Balance, and Glucose Homeostasis”, Pedroso et al., Nutrients, 2015
We thank the reviewer for his/her comment. Please see page 2, lines 65-73.
- The manuscript was not well-structured.
Part 5, “How high levels of BCAA influence the metabolic homeostasis of the body” should be involved in Part 2 “Adverse effects of isoleucine and valine”
We thank the reviewer for his/her suggestion. In agreement with his/her request, we included part 5 into part 2. Please see page 2 lines 74-92.
NAFLD is also a metabolic disease, Part 9 should be involved in Part 3
We thank the reviewer for his/her suggestion. In agreement with his/her request, we included part 9 into part 3. Please see page 4 lines 147-171.
Part 4 “Altered expression of BCAA metabolism genes”, it is improper that these contents presented as an independent part.
We thank the reviewer for his/her suggestion. However, the feeling of the authors is that part 4 is unique. If the reviewer thinks that it should be aggregated to part 3 (or any other parts of the manuscript), it is not a problem at all for the authors.
Part 7, 8 should be Part 6.2 and 6.3.
We thank the reviewer for his/her suggestion. In agreement with him/her, we included part 7 and 8 into part 5 (6 of the original version of the manuscript) as 5.2 and 5.3, respectively. Please see pages 6 and 7.
- The title should be revised. CVD and NAFLD were mentioned in the manuscript.
We thank the reviewer for his/her suggestion. The title has been revised.
Round 2
Reviewer 1 Report
This Reviewer consents with the publication of the paper in this revised form.